# The Role of Liver Stiffness Measurement and Spleen Stiffness Measurement in Predicting the Risk of Developing HCC

**DOI:** 10.3390/diagnostics14242867

**Published:** 2024-12-20

**Authors:** Rui Gaspar, Joana Mota, Maria João Almeida, Marco Silva, Guilherme Macedo

**Affiliations:** Gastroenterology and Hepatology, Centro Hospitalar de São João, 4200 Porto, Portugal; joanamfmota8@gmail.com (J.M.); maj.almeida.14@gmail.com (M.J.A.); marcocostasilva87@gmail.com (M.S.); guilhermemacedo59@gmail.com (G.M.)

**Keywords:** chronic liver disease, spleen elastrography, liver elastography, hepatocellular carcinoma, portal hypertension

## Abstract

Background/Objectives: Hepatocellular carcinoma (HCC) is the sixth most common cause of cancer worldwide. More than 90% of cases occur in cirrhotic patients, with the degree of fibrosis being the main risk factor for the development of HCC. Liver biopsy is the gold-standard for fibrosis assessment, but it is an invasive procedure. Liver stiffness measurement (LSM) has shown high accuracy for diagnosing liver cirrhosis, as well as for predicting decompensation and HCC development. More recently, spleen stiffness measurement (SSM) has presented excellent results for ruling in/out high-risk varices and the presence of clinical significant portal hypertension. The aim of our study was to evaluate the relationship between LSM and SSM and the risk of hepatocellular carcinoma. Methods: A prospective study on cirrhotic patients was performed in a tertiary center from January 2020 to May 2024. All patients were submitted to liver and spleen elastography (with a new probe of 100 Hz) by the same blinded operator and were treated in the same institution for the development of hepatocellular carcinoma. Results: We included 299 cirrhotic patients, 75.9% male, with a mean age of 61.8 years (±10.0). The median value of LSM was 25.7 kPa [4.5–75.0] and that of SSM was 44.6 kPa [7.9–100.0]. The median follow-up time was 505 days [114.0–1541.0]. During this period, 18 patients developed HCC, with a median time to HCC diagnosis after LSM and SSM of 321 days [63.0–1227.0]. LSM was the only factor associated with the development of HCC (*p* = 0.002) with an AUC of 0.715. On the other hand, SSM was not associated with the development of HCC. Conclusions: We found that the risk of developing HCC is associated with liver fibrosis but not with portal hypertension (assessed using SSM).

## 1. Introduction

Hepatocellular carcinoma (HCC) is the most common primary liver cancer, accounting for more than 85% of the cases. HCC is the 6th most common cause of cancer worldwide and the third leading cause of cancer-related death [1,2].

More than 90% of cases occur in a cirrhotic liver, and the main risk factors are chronic infection with hepatitis C virus (HCV) and hepatitis B virus (HBV), alcohol-associated liver disease, and metabolic dysfunction-associated steatotic liver disease (MASLD) [3,4,5]. The annual incidence of HCC in cirrhotic patients is 2–4% and it is estimated that almost one-third will develop liver cancer in their lifetime [1]. Diabetes mellitus, obesity, and smoking were also found to increase the risk for development of HCC [6,7].

HCC is a highly fatal cancer due to late diagnosis in the majority of the cases. Therefore, it is crucial to diagnose HCC in the earlier phases in order to improve the available treatments and prognosis [4].

The degree of liver fibrosis is the main risk factor for the development of HCC [8]. Thus, the assessment of liver fibrosis is of paramount importance to define high-risk groups who should be allocated for HCC screening.

Liver biopsy is the gold-standard for fibrosis assessment, but it is an invasive procedure and is associated with uncommon but serious complications [9].

Liver stiffness measurement (LSM) using transient elastography is a non-invasive, highly reproducible, and easy method for fibrosis evaluation and has become the standard of care for the evaluation of every patient with suspected liver disease [10,11]. LSM has shown high accuracy for staging liver disease and diagnosing liver cirrhosis, and it is one of the best non-invasive tools for predicting the presence of portal hypertension and risk of decompensation [12,13,14]. In addition, several studies have shown that LSM can be an important predictor of liver decompensation and HCC development in cirrhotic patients, with a systematic review finding that each unit increase in LSM will increase the risk of development of HCC by 11% [15].

In the last few years, spleen stiffness measurement (SSM) using spleen elastography with the new probe of 100 Hz has gained prominence and enhanced the results obtained using LSM alone, making it an important tool for the non-invasive evaluation of clinically significant portal hypertension. Existing data on SSM demonstrate excellent performance in ruling in/out high-risk varices. Furthermore, a study indicated that spleen stiffness could predict the risk of hepatocellular carcinoma after hepatitis C eradication with direct-acting antivirals [16,17,18].

The relationship between portal hypertension and HCC development remains a topic of debate. The pathogenesis of both portal hypertension and HCC is closely linked to chronic inflammation and angiogenesis (mainly associated with the increased expression of VEGF). Clinically significant portal hypertension is recognized as a predictive factor for HCC development, independent of the severity of cirrhosis [19,20]. However, some patients develop HCC without presenting portal hypertension.

Portal hypertension, along with surrogate markers such as esophageal or gastric varices, splenomegaly, or thrombocytopenia, is strongly associated with advanced liver disease—a significant risk factor for HCC. Nevertheless, there is no direct causal link between these markers and HCC development [12,19,21,22,23,24,25]. Most studies focus on the impact of clinically significant portal hypertension on the prognosis and recurrence risk of patients already diagnosed with HCC, rather than on the precise relationship between portal hypertension and the initial risk of HCC development [26,27,28]. Findings regarding the use of non-selective beta-blockers (NSBBs) and the risk of developing HCC are inconsistent. Some evidence suggests that NSBBs may reduce HCC risk, but this effect appears to be limited to patients with cirrhosis [29,30]. Additionally, a systematic review and meta-analysis concluded that TIPS (transjugular intrahepatic portosystemic shunt) is not associated with an increased risk of developing HCC [31].

The aim of our study was to evaluate the relationship between liver fibrosis and portal hypertension evaluated using LSM and SSM (with the new 100 Hz probe) and the risk of hepatocellular carcinoma.

## 2. Methods

### 2.1. Study Design, and Inclusion and Exclusion Criteria

A prospective study was conducted in the Gastroenterology and Hepatology Department of a tertiary centre from January 2020 to May 2024 on consecutive cirrhotic patients who were willing to participate in our study.

All patients underwent liver and spleen elastography at the same time and were prospectively treated in our Gastronterology and Hepatology department.

The diagnosis of cirrhosis was made based on liver biopsy or when liver stiffness measurement was higher than 12.5 kPa [32]. Patients with LSM < 12.5 kPa but with a previous histologic diagnosis of cirrhosis were also considered to participate in the study.

The study population comprised individuals with a history of cirrhosis, as evidenced using either clinical or histological criteria, who were at least 18 years of age and able to provide informed consent. Patients were excluded if they exhibited any of the following: non-cirrhotic portal hypertension, liver transplantation, a transjugular intrahepatic portosystemic shunt, acute or chronic portal vein thrombosis, hepatic congestion secondary to heart failure, splenectomy or the congenital absence of a spleen, pregnancy, a diagnosis of hepatocellular carcinoma before or at the time of LSM and SSM, the absence of relevant clinical data in their files, or an inability to undergo SSM (Figure 1).

### 2.2. Data Collection

The clinical and laboratory data were collected from electronic medical records of the hospital. Determining the etiology of cirrhosis required a detailed drug history, body mass index determination, HBV and HCV serologies, serum immunoglobulins, and a panel of autoantibodies for the diagnosis of autoimmune hepatitis (AIH) or primary biliary cholangitis (PBC), according to established criteria [33,34], as well as laboratory data for the diagnosis of hemochromatosis, Wilson’s disease, and α-1 antitrypsin deficiency [35,36,37]. HBV, DNA, and HCV RNA were requested when the antibodies were positive. The amount of alcohol consumption was calculated based on the description of the medium quantity and type of drinks consumed.

Clinical data collected included age, sex, race, a history of alcohol consumption, smoking, medication, and previous episodes of decompensation. Laboratory data included hemoglobin, white cell and platelet count (PLT), international normalized ratio (INR), albumin, aspartate aminotransferase (AST), alanine aminotransferase (ALT), gamma-glutamyl transferase (GGT), alkaline phosphatase (ALP), total and direct bilirubin, urea, creatinine, and alpha-fetoprotein (AFP). Scores from the model for end-stage liver disease (MELD) and Child–Pugh classification were calculated at the time of LSM and SSM and at the time of the diagnosis of hepatocellular carcinoma (HCC).

Hepatocellular carcinoma (HCC) screening was performed every 6 months or at admission by conducting abdominal ultrasound and AFP [3]. In cases of HCC suspicion, an abdominal magnetic resonance imaging (MRI) scan or an abdominal computed tomography scan was performed to establish the diagnosis. In doubtful cases, a liver biopsy of the nodule was taken. All HCC cases were discussed by a multidisciplinary team, where all available images were evaluated by an experienced team that included hepato-biliary surgeons, hepatologists, radiologists, and oncologists.

All laboratory work-ups and imaging studies, when needed, were performed at our institution.

### 2.3. Transient Elastography Examination (LSM and SSM)

The SSM and LSM were performed at the same time with patients in the supine position, with both arms in maximum abduction, and after at least 6 h of fasting, using the FibroScan Expert 630 (Echosens®, Paris, France) device. SSM and LSM were performed by a single experienced operator.

LSM was performed with M or XL probes depending on the thickness of subcutaneous fat, while SSM was performed using a 100 Hz M probe alone. Spleen marking was performed with an ultrasound probe included in the same device before SSM.

For each patient, 2 separate sets of 10 measurements for each were taken for LSM and SSM, and the average value was recorded. Quality criteria applied for SSM measurement were similar to those for LSM (≥60% success rate; IQR < 30% of median).

### 2.4. Statistical Analysis

The data were analyzed using SPSS version 27.0 (IBM Corp, Armonk, NY, USA). The normality of variables was assessed using histograms and the Shapiro–Wilk test. Continuous variables are presented as the mean ± standard deviation when normally distributed, or as the median (range) when they deviate from normality. Categorical variables are reported as absolute numbers (*n*) and relative frequencies (%). For categorical variables, the difference between groups will be tested using the chi-square test or Fisher’s exact test, as appropriate. Group comparisons for categorical variables were tested using the X^2^-test or Fisher’s exact test, as appropriate. Comparisons of continuous variables between groups were conducted using either the *t*-test or the Mann–Whitney U test, depending on whether the normality assumption was met.

Sensitivity, specificity, positive and negative predictive values, positive and negative likelihood ratios, and diagnostic accuracy were calculated based on the ROC curve analysis. To determine the optimal cut-off value for LSM in identifying patients who developed HCC, the ROC curve was constructed, and the cut-off value was established using Youden’s index.

### 2.5. Ethical Considerations

This study was conducted in accordance with the Declaration of Helsinki and informed consent was obtained from all participants in the study. The study was approved by the Ethics Committee of Centro Hospitalar São João.

## 3. Results

### 3.1. Study Population

During the study period (January 2020–May 2024), 344 patients were enrolled. Among the 344 patients, 20 were excluded due to an HCC diagnosis at the same time of performing LSM and SSM or in the 6 months prior; 17 due to missing relevant clinical data; 8 due to other reasons; 5 due to a failed SSM examination because of obesity (BMI > 35) and large volume ascites; 1 due to previous splenectomy; and 2 due to liver congestion secondary to heart failure. The final number of patients included was 299 (Figure 1).

The mean age was 61.8 years (±10.0) and 75.9% were male.

The diagnosis of cirrhosis was based on liver elastography in the 73.6% of patients, and the most common etiologies were alcohol-associated liver disease (66.9%), MASLD (17.1%), and HCV (9.7%). All HCV patients had a sustained virological response at enrollment.

Almost one-third of the patients (30.4%) had active alcohol consumption and 44.1% reported previous alcohol consumption. Sixty-four patients (21.4%) were active smokers. The median MELD score was 7.3 [6.0–22.0]. Most of the patients were classified as Child A (77.6%), 20.4% were classified as Child B, and 2.0% were classified as Child C. Seventy-six patients (25.4%) had previous episodes of cirrhosis decompensation.

During the period of follow-up, there were two patients transplanted and four died due to liver-related complication.

Clinical characteristics are displayed in Table 1 and laboratory data in Table 2.

### 3.2. LSM and SSM

The median value of LSM was 25.7 kPa [4.5–75.0] and that of SSM was 44.6 kPa [7.9–100.0]. LSM and SSM are depicted in Table 3a.

### 3.3. Hepatocellular Carcinoma

The median time of follow-up was 505 days [114.0–1541.0]. During the period of analysis, 18 patients (6.0%) developed HCC. The median time to HCC diagnosis after LSM and SSM was 321 days [63.0–1227.0]. Regarding the patients who developed HCC, there were 17 males (94.4%) and the mean age was 60.8 years (±9.4). The median value of LSM was 51.6 kPa [17.1–75.0] and that of SSM was 63.8 kPa [22.3–100.0], as shown in Table 3b.

The main cause of liver disease was alcohol (83.3%) and only 33% had previous episodes of decompensation.

The majority of patients were classified at admission as Child–Pugh A (55.6%), 33.3% were Child–Pugh B, and 11.1% of the cases were Child C.

The cumulative incidence rates of HCC were 0.67% at 6 months, 3.3% at 1 year, 5.4% at 2 years and 6.0% at 3.5 years

### 3.4. LSM, SSM, and Other Scores for Prediction of Development of HCC

LSM was statistically associated with the development of HCC in our population of cirrhotic patients (*p* = 0.002). On the other hand, there were no statistically significant differences between the development of HCC and SSM, APRI, FIB-4, MELD, or ALBI scores.

Table 4 depicts the value of each predictor in patients who did or did not develop HCC. 

Figure 2 depicts the ROC curve produced using LSM, which achieved an area under the curve (AUC) of 0.715 (95% confidence interval [CI]:0612–0.819), demonstrating only fair capacity for predicting the development of HCC. The optimal cut-off was determined as the point on the ROC curve nearest to the upper left corner. For the prediction of HCC development, LSM > 24.55 kPa had a sensitivity of 83.3% and a specificity of 49.1%, *p* = 0.002.

## 4. Discussion

HCC is the one of the most common causes of cancer worldwide and is one of the most fatal [1,2]. In the last few years, there have been significant improvements in all selected treatments for HCC: better surgical techniques (for surgical resection and liver transplant), the better selection of patients for liver transplant with reduced rates of HCC recurrence, the higher efficacy of thermal ablation and chemoembolization techniques, and the emergence of a wide variety of systemic therapies with much higher survival rates compared to sorafenib [4]. However, survival rates have not increased correspondingly, as prognosis is intrinsically associated with an earlier diagnosis.

Liver fibrosis is the main risk factor for the development of HCC, with more than 90% of cases occurring in patients with cirrhosis [3,8]. Due to the invasiveness and risks associated with liver biopsy, LSM using transient elastography has become the cornerstone for liver fibrosis evaluation, presenting an AUROC of 0.94 (95% CI: 0.93–0.95) for the diagnosis of cirrhosis and eliminating the need of liver biopsy in 90% of the cases [38,39,40].

In our study, we evaluated the capacity of non-invasive tools and scores to predict the development of HCC. Consistent with previous studies, we did not find any relationship with the MELD score, the ALBI score, and the development of HCC. The MELD score was validated for the prediction of survival in patients with cirrhosis and for prioritizing liver transplant allocation, but not for predicting the risk of developing HCC. Similarly, the ALBI score was used as a prognostic factor in HCC patients and not to predict its development [41,42]. Akin to APRI e FIB-4, some comparative studies have evaluated their ability to predict the development of HCC. However, the accuracy is low, and most studies have only included HBV, HCV, or MASLD patients, limiting the data for other liver disease etiologies [43,44,45,46]. In our study, which includes a large population with different liver disease etiologies, APRI and FIB-4 did not predict the development of HCC.

LSM using transient elastography is considered the most adequate non-invasive tool for evaluating liver fibrosis. There are also studies reporting a direct association between LSM in cirrhotic patients and predicting cirrhosis-related complications, including decompensation and death [47,48,49]. In recent years, several studies reported the association between LSM and the risk of developing HCC [46,50,51]. Initial studies compared patients with high-grade fibrosis to those with low-grade fibrosis, reporting that patients with LSM > 8 or LSM > 10 had a higher risk of developing HCC. More recent studies have almost focused solely on patients with cirrhosis [51,52,53,54]. These studies showed that, in cirrhotic patients, higher values of LSM were associated with a higher risk of developing HCC (>24 kPa, >38.5 kPa, and >28.7 kPa) [39,47,55]. In our study, which only included cirrhotic patients, we found that higher values of LSM were associated with a higher risk of developing HCC. The cut-off of 24.55 kPa (sensitivity of 83.3% and specificity of 49.1%) identified patients with a higher probability of developing HCC. This fact is intrinsically linked to the molecular mechanisms responsible for the development of HCC; activated hepatic stellate cells promote the synthesis of extracellular matrix components, which increase growth factors. These growth factors create a pro-tumorigenic microenvironment, causing genetic alterations, chromosomal instabilities, the activation of oncogenic pathways, and reduced tumor surveillance using natural killer cells [56,57]. As the grade of fibrosis increases, these mechanisms become more common, leading to a higher risk of developing HCC.

Previously, the determination of portal hypertension was performed using HVPG measurement, a procedure that requires expertise and is costly and invasive [58]. More recently, spleen stiffness measurement (SSM) has shown excellent correlation with the degree of PH, demonstrating excellent performance in predicting the presence of high-risk varices [17,59,60]. Therefore, SSM is now being used as a non-invasive tool for evaluating portal hypertension.

The correlation between portal hypertension and the risk of developing HCC has never been extensively studied. Some studies suggest a theoretical relationship due to altered sinusoidal perfusion and local ischemic condition, which could favor the progression of fibrosis and the carcinogenesis process [46,61]. There is only one study suggesting that spleen stiffness could predict the risk of late recurrence of HCC after resection, but, as far as we know, this is the first study to evaluate the correlation between portal hypertension, evaluated using SSM, and development of HCC [62].

Nevertheless, we were unable to find a significant association between portal hypertension, evaluated using SSM, and the risk of developing HCC. This suggests that fibrosis is more important for the carcinogenesis pathway than the presence of portal hypertension.

To the best of our knowledge, and as one of the major strengths of our work, this is the first study to evaluate the relationship between portal hypertension, evaluated using SSM, and the development of HCC. Additionally, we assessed the relationship between liver fibrosis (evaluated using non-invasive scores and LSM) and the risk of developing HCC. Another positive aspect is that we included a large number of cirrhotic patients, all of whom were treated in accordance with international guidelines, i.e., they underwent ultrasound and alpha-fetoprotein measurement every 6 months. All patients diagnosed with HCC were also discussed in multidisciplinary team meetings to ensure accurate diagnoses. Furthermore, we included patients with multiples etiologies and stages of liver disease.

This study has significant limitations. Firstly, it is a single-center study, which could limit the external validity of our results. Secondly, although spleen elastography is becoming an important tool for portal hypertension evaluation, it is not yet the gold-standard. Thirdly, although our median follow-up is 505 days, it could be considered a relatively short follow-up period for the development of HCC, limiting the robustness of our conclusions. Fourthly, although we included patients with different etiologies, viral hepatitis, known to present higher risk of developing HCC, are underrepresented in our population. This is consistent with trends in most developed countries, where alcohol-related liver disease and MASLD are the primary causes. Finally, there were only 18 cases of HCC during the follow-up, which may affect the statistical strength and limit our conclusions.

## 5. Conclusions

Identifying patients at increased risk of developing HCC is one of the main goals in the evaluation of cirrhotic patients.

In our study, consistent with all the molecular mechanisms previously shown to increase liver carcinogenesis, we found that the risk of developing HCC is associated with liver fibrosis, identifying a cut-off value of 24.55 kPa for LSM, with a sensitivity of 83.3% for predicting HCC development. On the other hand, we did not observe an association between HCC development and the presence of portal hypertension, as assessed using SSM.

## Figures and Tables

**Figure 1 diagnostics-14-02867-f001:**
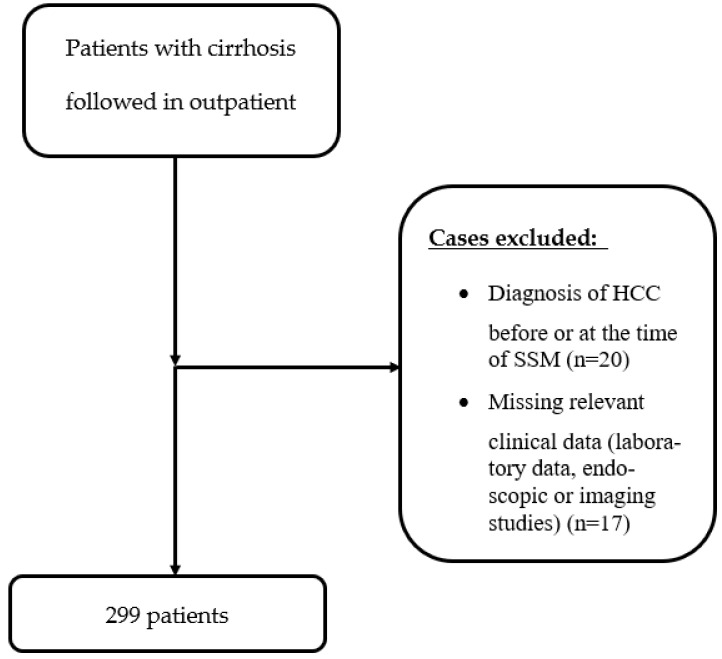
Study flowchart.

**Figure 2 diagnostics-14-02867-f002:**
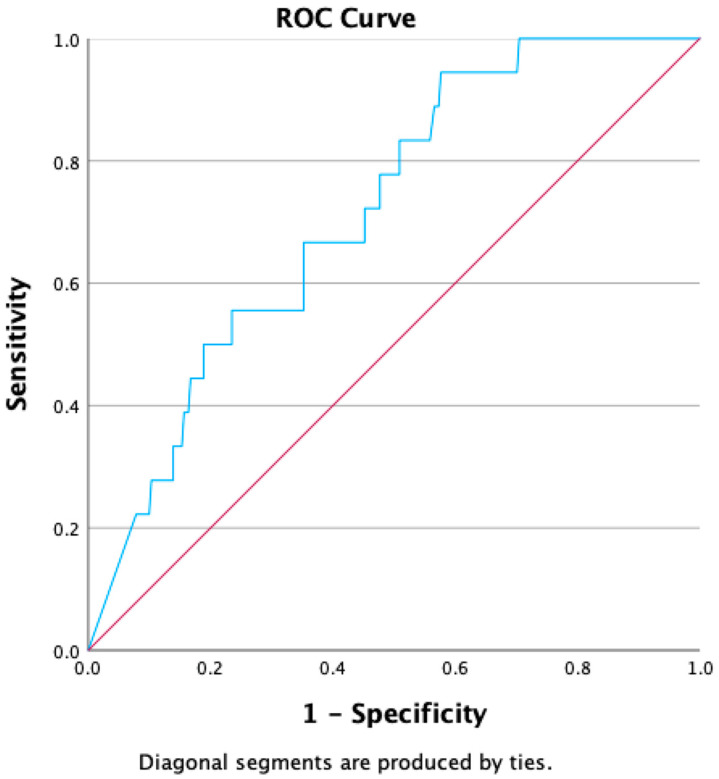
AUC of LSM for the prediction of development of HCC (in blue); In red the line of no discrimination.

**Table 1 diagnostics-14-02867-t001:** Characteristics of the 299 cirrhotic patients.

	*n* = 299
**Mean age (SD)**	61.8 years (±10.0)
**Male sex**	227 (75.9%)
**Cirrhosis diagnosis:** LSM Histological	220 (73.6%) 79 (26.4%)
**Cirrhosis etiology:** Alcohol MASLD HCV HBV Autoimmune hepatitis Hemochromatosis Drug-induced liver injury Primary biliary cholangitis Alpha-1 antitrypsin deficiency Wilson’s disease	200 (66.9%) 51 (17.1%) 29 (9.7%) 6 (2.0%) 5 (1.7%) 3 (1.0%) 2 (0.7%) 1 (0.3%) 1 (0.3%) 1 (0.3%)

Legend: HBV: hepatitis B virus; HCV: hepatitis C virus; LSM: liver stiffness measurement; MASLD: metabolic dysfunction-associated steatotic liver disease; SD: Standard deviation.

**Table 2 diagnostics-14-02867-t002:** Laboratory data.

	*n* = 299
Hemoglobin (g/dL)	13.8 [7.5–17.9]
Platelets (×10^9^ L)	136.5 [35.0–416.0]
Albumin (g/L)	40.4 [19.1–52.8]
AST (U/L)	43.5 [15.0–281.0]
ALT (U/L)	34.0 [10.0–240.0]
GGT (U/L)	104.0 [13.0–1987.0]
Total bilirubin (mg/dL)	0.97 [0.3–33.5]
Creatinine (mg/dL)	0.75 [0.3–7.6]
Alpha-fetoprotein (ng/mL)	6.0 [0.9–224]

Legend: ALT: alanine aminotransferase; AST: aspartate aminotransferase; GGT: gamma-glutamyl transferase.

**Table 3 diagnostics-14-02867-t003:** (**a**) LSM and SSM of all the cirrhotic patients. (**b**) LSM and SSM of the patients with HCC.

(**a**)	
	*n* = 299
LSM (kPa)	25.7 [4.5–75.0]
IQR (%)	15% [0.0–30.0]
CAP (dB/m)	254.5 [100.0–394.0]
IQR (dB/m)	IQR 30.0 [0–78.0]
SSM (kPa)	44.6 [7.9–100]
IQR (kPa)	6.5 [0–39.0]
(**b**)	
	*n* = 18
LSM (kPa)	51.6 [17.1–75.0]
IQR (%)	17% [0.0–30.0]
CAP (dB/m)	234.5 [100.0–382.0]
IQR (dB/m)	IQR 27.0 [2.0–65.0]

Legend: CAP: controlled attenuation parameter; IQR: inter-quartile range; LSM: liver stiffness measurement; SSM: spleen stiffness measurement.

**Table 4 diagnostics-14-02867-t004:** Scores in patients who did or did not develop HCC.

Characteristic	Overall *n*= 299 ^1^	HCC *n* = 18 ^1^	No HCC *n* = 281 ^1^	*p*-Value
LSM	25.7 [16.3, 49.0]	51.55 [25.65, 72.9]	25.3 [15.7, 46.2]	0.002
SSM	44.6 [30.7, 68.2]	63.8 [32.7, 90.9]	44.2 [30.2, 67.2]	0.077
MELD	7.28 [6.43, 9.48]	8.01 [6.59, 11.48]	7.24 [6.43, 9.38]	0.154
Child	5.0 [5.0, 6.0]	5.0 [5.0, 7.3]	5.0 [5.0, 6.0]	0.132
APRI	0.87 [0.53, 1.56]	0.90 [0.46, 3.19]	0.87 [0.54, 1.55]	0.200
FIB-4	3.41 [2.11, 5.85]	3.94 [2.01, 8.67]	3.4 [2.12, 5.72]	0.275
ALBI	−3.43 [−3.66, −2.94]	−2.92 [−3.66, −2.44]	−3.44 [−3.66, −2.97]	0.075

^1^ Median (Q1 and Q3). Legend: ALBI: albumin–bilirubin; APRI: AST-to-platelet ratio index; FIB-4: fibrosis-4 index; MELD: model for end-stage liver disease; LSM: liver stiffness measurement; SSM: spleen stiffness measurement.

## Data Availability

The original contributions presented in this study are included in the article. Further inquiries can be directed to the corresponding author.

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
