# Peer review of "The Role of Liver Stiffness Measurement and Spleen Stiffness Measurement in Predicting the Risk of Developing HCC"

_diagnostics, 2024, doi:10.3390/diagnostics14242867_

Round 1

Reviewer 1 Report (Previous Reviewer 2)

Comments and Suggestions for Authors

Thank you for adressing my comments

Author Response

Thank you for your comments.

Best regards

Reviewer 2 Report (New Reviewer)

Comments and Suggestions for Authors

In this manuscript, the authors evaluated the predictive value of liver and spleen stiffness for the development of HCC in a large prospective cohort of patients with cirrhosis. The authors found that spleen stiffness was not associated with the risk of developing HCC, while liver stiffness was significantly associated with the development of HCC. The study is interesting however there are some points that could be improved.

·         The authors analyzed the relationship between the development of HCC, LSM SSM and other scores (including MELD, FIB-4, etc.), but the data were not shown. The authors need to include a table showing the value of each predictor in patients who did or did not develop HCC.

·         Are only baseline LSM and SSM available? Did the authors have data on the predictive value of changes from baseline in liver and spleen stiffness?

·         Did the author use the Cox proportional hazards regression model to calculate the risk of HCC according to baseline LSM and SSM? In addition, if a significant number of deaths or liver transplants occurred during follow-up (please include this data in the description of the population), a competing risk model could be used (e.g. Fine and Gray).

Minor

·         Check the language throughout the manuscript (eg. verb tenses are not consistent in the statical methods section).

·         Please review terminology when referring to alcohol-associated liver disease according to the latest recommendation (10.1016/j.aohep.2023.101143).

·         I suggest reducing the number of references and keeping only the most relevant ones.

Author Response

Porto, 12th December, 2024

Dear Editor,

We kindly thank you for the opportunity to submit the third revised version of the manuscript " Role of liver stiffness measurement and spleen stiffness measurement in predicting the risk of HCC " (diagnostics-3351988)

We found the comments very thoughtful and tried to address all of the points raised.

We now hope that after this effort, you consider the current manuscript acceptable for publication.

On behalf of all the co-authors, we are looking forward to hearing from you.

In this manuscript, the authors evaluated the predictive value of liver and spleen stiffness for the development of HCC in a large prospective cohort of patients with cirrhosis. The authors found that spleen stiffness was not associated with the risk of developing HCC, while liver stiffness was significantly associated with the development of HCC. The study is interesting however there are some points that could be improved. 

  • The authors analyzed the relationship between the development of HCC, LSM SSM and other scores (including MELD, FIB-4, etc.), but the data were not shown. The authors need to include a table showing the value of each predictor in patients who did or did not develop HCC. 

We thank the Reviewer for the comments. We created a table as suggested an included in the text.

Characteristic

Overall 
N = 2991

HCC
N = 181

No HCC
N = 2811

p-value

LSM

25.7 [16.3, 49.0]

51.55 [25.65, 72.9]

25.3 [15.7, 46.2]

0.002

SSM

44.6 [30.7, 68.2]

63.8 [32.7, 90.9]

44.2 [30.2, 67.2]

0.077

MELD

7.28 [6.43, 9.48]

8.01 [6.59, 11.48]

7.24 [6.43, 9.38]

0.154

Child

5.0 [5.0, 6.0]

5.0 [5.0, 7.3]

5.0 [5.0, 6.0]

0.132

APRI

0.87[0.53, 1.56]

0.90 [0.46, 3.19]

0.87 [0.54, 1.55]

0.002

FIB-4

3.41 [2.11, 5.85]

3.94 [2.01, 8.67]

3.4 [2.12, 5.72]

0.275

ALBI

-3.43 [-3.66, -2.94]

-2.92 [-3.66, -2.44]

-3.44 [-3.66, -2.97]

0.075

1Median (Q1, Q3)

  • Are only baseline LSM and SSM available? Did the authors have data on the predictive value of changes from baseline in liver and spleen stiffness?

      We thank the Reviewer for the comments. As we included only cirrhotic patients, we only have the baseline values. Although, as a good suggestion, we will try to do seriated evaluations to include in the next studies.

  • Did the author use the Cox proportional hazards regression model to calculate the risk of HCC according to baseline LSM and SSM? In addition, if a significant number of deaths or liver transplants occurred during follow-up (please include this data in the description of the population), a competing risk model could be used (e.g. Fine and Gray). 

      We thank the Reviewer for the comments. A multivariate analysis was conducted using the Cox proportional hazards model, with HCC as the dependent variable and only LSM (p=0.004) was significantly associated. During the period of follow-up there were 2 patients transplanted and 4 died due to liver-related complications. Due to this reduced number, we did not perform a competing risk model.

Minor

  • Check the language throughout the manuscript (eg. verb tenses are not consistent in the statical methods section). 

      We thank the Reviewer for the comments. The changes have been made accordingly

  • Please review terminology when referring to alcohol-associated liver disease according to the latest recommendation (10.1016/j.aohep.2023.101143).

      We thank the Reviewer for the comments. The changes have been made accordingly

  • I suggest reducing the number of references and keeping only the most relevant ones.

We thank the Reviewer for the comments. In the first revision, the Academic Editor asked to increase the number of references and after our revision he was satisfied with the increase number of references. That is the reason of the high number of references.

Round 2

Reviewer 2 Report (New Reviewer)

Comments and Suggestions for Authors

I would like to thank the authors for responding to my comments. There is only one additional point to be clarified. Please note that Table 4 is not quoted in the main text, and furthermore, contrary to what is mentioned in the main text, Table 4 shows that the APRI score is significantly different between HCC and non-HCC patients. Please comment on such difference. 

Author Response

Dear reviewer

Porto, 16th December, 2024

We kindly thank you for the opportunity to submit a revised version of the manuscript "Role of liver stiffness measurement and spleen stiffness measurement in predicting the risk of HCC?" (Diagnostics-3225577)

We found the comments very instructive and we answered to the questions.

We hope that you consider the current manuscript acceptable for publication.

On behalf of all the co-authors, we are looking forward to hearing from you.

Yours sincerely,

Rui Gaspar

Reviewers' Comments to Author:

We thank the Reviewer for the comments. In fact, it was a spelling mistake (0.200 instead of 0.002), and that is why it is not on bold. We also mentioned table 4 in the text and also completed its legend.

This manuscript is a resubmission of an earlier submission. The following is a list of the peer review reports and author responses from that submission.

Round 1

Reviewer 1 Report

Comments and Suggestions for Authors

I appreciate the work done by your team for the development of this manuscript.

I carefully read the manuscript sent for publication and I want to convey to you some deficiencies that caught my attention and which I will expose in the following.

Of course, the attempt to correlate the risk of developing HCC with portal hypertension is a commendable initiative to the extent that portal hypertension would develop independently of the liver fibrosis process.

In addition, if portal hypertension would appear independently of the liver fibrosis process, possibly before an advanced fibrosis process, then the explanation should be sought elsewhere, that is, in the production in the portal circulation under increased pressure of some substances capable of induce the process of carcinogenesis.

But, both in medical practice and in specialized medical literature, it has been proven that there is a large number of patients who, although they are in the last stage of liver fibrosis, do not have signs and symptoms of portal hypertension.

For this type of patients, the predictability of portal hypertension for the development of HCC does not stand.

Also, there are patients in whom the fibrosis is not at a very advanced stage, but develop HCC. Or, in these patients it cannot even be a question of portal hypertension, so that even in these patients we cannot speak of a relationship between portal hypertension and HCC.

The introduction is very long, it evokes many data already known in the specialized literature and focuses too little on the problem addressed in the manuscript.

In principle, the study is well conducted, the group of patients chosen is large enough for statistical significance, the statistical methods are chosen appropriately.

The obtained results are represented adequately and intelligibly.

The conclusions can be significantly improved, especially since the discussion chapter mentions the phrases: "This fact is intrinsically linked to the molecular mechanisms responsible for the development of HCC: activated hepatic stellate cell promote synthesis of extracellular matrix components, which increase growth factors. These growth factors create a pro-tumorigenic microenvironment, causing genetic alterations, chromosomal instabilities, activation of oncogenic pathways and reduced tumor surveillance by natural killer cells. As the degree of fibrosis increases, these mechanisms become more common, leading to a higher risk of developing HCC. "

Starting from the sentences above, in fact, the failure to demonstrate that portal hypertension is a predictive factor for HCC was already known.

Under these conditions, knowing that research failures are also published in the literature, I can say that this failure was published, but it was an already known failure.

The bibliography is correctly inserted in the text of the manuscript, and is relevant to the topic addressed, but it is very old, very few bibliographic references go beyond the years 2019-2020

In addition, a similarity ratio of 40% given by iThenticate is too high for the manuscript to be published.

Author Response

Porto, 18th November, 2024

Dear Editor,

We kindly thank you for the opportunity to submit a revised version of the manuscript "Does portal hypertension increase the risk of HCC?" (Diagnostics-3225577)

We found the comments very thoughtful and instructive and we have addressed all of the points raised.

In the manuscript, the revisions are in red font.

We hope that you consider the current manuscript acceptable for publication.

On behalf of all the co-authors, we are looking forward to hearing from you.

Yours sincerely,

Rui Gaspar

Reviewers' Comments to Author:

I appreciate the work done by your team for the development of this manuscript.

I carefully read the manuscript sent for publication and I want to convey to you some deficiencies that caught my attention and which I will expose in the following.

Of course, the attempt to correlate the risk of developing HCC with portal hypertension is a commendable initiative to the extent that portal hypertension would develop independently of the liver fibrosis process.

We thank the Reviewer for the comments.

In addition, if portal hypertension would appear independently of the liver fibrosis process, possibly before an advanced fibrosis process, then the explanation should be sought elsewhere, that is, in the production in the portal circulation under increased pressure of some substances capable of induce the process of carcinogenesis.

But, both in medical practice and in specialized medical literature, it has been proven that there is a large number of patients who, although they are in the last stage of liver fibrosis, do not have signs and symptoms of portal hypertension.

For this type of patients, the predictability of portal hypertension for the development of HCC does not stand. Also, there are patients in whom the fibrosis is not at a very advanced stage, but develop HCC. Or, in these patients it cannot even be a question of portal hypertension, so that even in these patients we cannot speak of a relationship between portal hypertension and HCC.

We thank the Reviewer for the comments. In fact, there are some patients that develop portal hypertension without advanced fibrosis, but normally associated with portal vein thrombosis, and they do not have higher risk of HCC.

On the other hand, a lot of patients do not develop portal hypertension and other patients without advanced fibrosis develop HCC, reinforcing that the portal hypertension may not be a risk factor for HCC. This is one of the most important reasons to perform our study.

The introduction is very long, it evokes many data already known in the specialized literature and focuses too little on the problem addressed in the manuscript. In principle, the study is well conducted, the group of patients chosen is large enough for statistical significance, the statistical methods are chosen appropriately. The obtained results are represented adequately and intelligibly.

We thank the Reviewer for the comments. We reduced some data already known and add some information regarding portal hypertension and the risk of HCC, as well as we clarified the main aim of this manuscript. After your suggestion, we think that we significantly improved the introduction.

The conclusions can be significantly improved, especially since the discussion chapter mentions the phrases: "This fact is intrinsically linked to the molecular mechanisms responsible for the development of HCC: activated hepatic stellate cell promote synthesis of extracellular matrix components, which increase growth factors. These growth factors create a pro-tumorigenic microenvironment, causing genetic alterations, chromosomal instabilities, activation of oncogenic pathways and reduced tumor surveillance by natural killer cells. As the degree of fibrosis increases, these mechanisms become more common, leading to a higher risk of developing HCC. "

We thank the Reviewer for the comments.

As suggested, we clarified the conclusion to give more value to our manuscript.

The changes have been made accordingly

Starting from the sentences above, in fact, the failure to demonstrate that portal hypertension is a predictive factor for HCC was already known. Under these conditions, knowing that research failures are also published in the literature, I can say that this failure was published, but it was an already known failure.

We thank the Reviewer for the comments.

Although some reports that do not show relationship between HCC and portal hypertension, this matter is still debatable. Our study has the advantage of using a very recent technology for assessment and graduation of portal hypertension, facilitating the evaluation of our patients without the need of HVPG measurements.

The bibliography is correctly inserted in the text of the manuscript, and is relevant to the topic addressed, but it is very old, very few bibliographic references go beyond the years 2019-2020

We thank the Reviewer for the comments. There are no recent papers, as most of them report the risk of decompensation after HCC treatment and not the relationship between portal hypertension and development of HCC. However, we introduced some more recent papers that investigated the relation between portal hypertension and risk of recurrence of HCC.

In addition, a similarity ratio of 40% given by iThenticate is too high for the manuscript to be published.

We thank the Reviewer for the comments. Most of the cases are due to similarities in the methodology published in another paper by our group. Anyway, we made significant changes to the text in order to reduce this percentage of similarity.

Reviewer #2: Several ethiologies to cirrhosis dispose to hepatocellular carcinoma (HCC) and for example, HBV has a high degree of carcinogenicity. Cirrhosis is often complicated by portal hypertension (PH), which per se also may induce HCC. In the present study, the authors aimed to evaluate the relationship between liver and spleen stiffness measurements as an expression for PH and the risk of HCC.

Although of interrest this reviewer has some comments and questions for the authors consideration. 

  1. The most common ethiologies of cirrhosis in this study was alcohol followed by MASLD. The most carcinogenic ethiologies were spearsely represented, which may affect the results. Please comment.

We thank the Reviewer for the comments.

In our population, most cases are caused by alcohol and MASLD and as a developed country most of our population is vaccinated to HBV and due to the use of DAA the cases of cirrhosis due to HCV are also decreasing. However, as we know that patients with chronic liver disease due to viral hepatitis have higher risk of HCC, we add this information to the limitations of our study.

  1. Did you find any indication whether the risk of HCC was different within the different patients group according to ethiology?

We thank the Reviewer for the comment.

We did not find any difference, although we only have 18 cases of HCC.

  1. Of 344 recruited patients, only 18 patients had HCC, a relationship that may affect the statistical strength, wherefore the results should be interpreted with cautiousness.

We thank the Reviewer for the comments. We totally agree with your comment and add that information to the limitations of the study. The changes have been made accordingly.

  1. The gold standard of measuring portal pressure is by the difference between the wedged and free hepatic venous pressures as the hepatic venous pressure gradient. Did you have the opportunity to compare gold standard measures with your liver/spleen stiffness measures? Did you find any correlations?  

We thank the Reviewer for the question.

We did not compare directly as it is not easy to perform HVPG measurements in our center. A lot of studies have already shown a direct association between spleen stiffness (as a non-invasive evaluation of portal hypertension) and the presence of high risk varices.

The direct comparison between spleen stiffness measurement and HVPG measurement is more difficult and also lacks total validation, but there are already some studies that compared both techniques - (Lancet Gastroenterol Hepatol. 2024 Dec;9(12):1111-1120.; doi: 10.1016/S2468-1253(24)00234-6. Epub 2024 Sep 23.)

  1. Please provide details on the coefficients of variation of your stiffness measurements.

We thank the Reviewer for the comment. The IQR of spleen stiffness measurement is not defined in any paper or guideline, but it is recommended to follow the maximum of 30% of IQR as a parameter of quality.

We found a median IQR of 6.5 kPa, with a minimum of 0 and a maximum of 39.0.

  1. Are you aware of other studies that support your data?

We thank the Reviewer for the comment. There are no more studies, nor supporting, nor refusing our data. This was one of the main reasons to perform this study. There are a lot of reports showing a direct relation between liver fibrosis and risk of HCC but not with spleen stiffness.

There is one paper that evaluated the role of liver and spleen stiffness in the recurrence of HCC after resection but motivated several criticisms in the world of Hepatology - Role of liver and spleen stiffness in predicting the recurrence of hepatocellular carcinoma after resection, J Hepatol. 2019 Mar;70(3):440-448.doi:10.1016/j.jhep.2018.10.022. Epub 2018 Oct 31 and a response letter: Predicting post-resection recurrence of hepatocellular carcinoma: Spleen stiffness vs. ALBI grade, Journal of Hepatology 2019 vol. 70 j 788–816

  1. Gold standard of assessing the amount of fibrosis is by liver biopsy. Were you able to compared results of liver biopsies with liver stiffness measurements in a subset of you patients?

We thank the Reviewer for the question.

We made this several years before and also compared CAP measurements with steatosis in liver biopsy - GE Port J Gastroenterol 2017 Jul;24(4):161-168.

 doi: 10.1159/000453364. Epub 2016 Dec 23. Diagnostic Accuracy of Controlled Attenuation Parameter for Detecting Hepatic Steatosis in Patients with Chronic Liver Disease

  1. Since liver stiffness not solely reflect fibrosis and since fibrosis does not express the degree of PH I find the title misleading and it should be modified.

We thank the Reviewer for the comment. We agree and changed the name to: "Role of liver stiffness measurement and spleen stiffness measurement in predicting the risk of HCC"

  1. MAFLD is now termed MASLD. Please correct. 

We thank the Reviewer for the comment.

The changes have been made accordingly.

Reviewer 2 Report

Comments and Suggestions for Authors

Several ethiologies to cirrhosis dispose to hepatocellular carcinoma (HCC) and for example, HBV has a high degree of carcinogenicity. Cirrhosis is often complicated by portal hypertension (PH), which per se also may induce HCC. In the present study, the authors aimed to evaluate the relationship between liver and spleen stiffness measurements as an expression for PH and the risk of HCC.

Although of interrest this reviewer has some comments and questions for the authors consideration. 

1. The most common ethiologies of cirrhosis in this study was alcohol followed by MASLD. The most carcinogenic ethiologies were spearsely represented, which may affect the results. Please comment.

2. Did you find any indication whether the risk of HCC was different within the different patients group according to ethiology?

3. Of 344 recruited patients, only 18 patients had HCC, a relationship that may affect the statistical strength, wherefore the results should be interpreted with cautiousness.

4. The gold standard of measuring portal pressure is by the difference between the wedged and free hepatic venous pressures as the hepatic venous pressure gradient. Did you have the opportunity to compare gold standard measures with your liver/spleen stiffness measures? Did you find any correlations?  

5. Please provide details on the coefficients of variation of your stiffness measuremets.

6. Are you aware of other studies that support your data?

7. Gold standard of assessing the amount of fibrosis is by liver biopsy. Were you able to compared results of liver biopsies with liver stiffness measurements in a subset of you patients?

8. Since liver stiffness not solely reflect fibrosis and since fibrosis does not express the degree of PH I find the title misleading and it should be modified.

9. MAFLD is now termed MASLD. Please correct. 

Comments on the Quality of English Language

-

Author Response

(The authors gave the same response as above.)
